# Psychological Resilience and Coping Strategies with Anxiety among Malaysian Medical Students during the COVID-19 Pandemic

**DOI:** 10.3390/ijerph20031894

**Published:** 2023-01-19

**Authors:** Bentham Liang Sen Teh, Jin Kiat Ang, Eugene Boon Yau Koh, Nicholas Tze Ping Pang

**Affiliations:** 1Department of Psychiatry, Faculty of Medicine and Health Sciences, University Putra Malaysia, Selangor 43400, Malaysia; 2Faculty of Medicine and Health Sciences, University Malaysia Sabah, Kota Kinabalu 88400, Malaysia

**Keywords:** COVID-19, medical students, anxiety, self-esteem, self-efficacy, coping

## Abstract

Coronavirus Disease 2019 (COVID-19) swept the world by storm and caused a myriad of devastating consequences, particularly disruptions in medical education. This study aims to examine the association between sociodemographic factors, psychological factors, coping strategies and anxiety among medical students, as well as to identify the predictors of anxiety among them. A cross-sectional study design was used. Self-rated Rosenberg Self-Esteem Scale (RSES), General Self-Efficacy Scale (GSES), Brief Coping Orientation to Problems Experienced Scale (Brief COPE), and General Anxiety Disorder-7 Scale (GAD-7) were used. A total of 371 respondents from a tertiary education center were recruited. The prevalence of anxiety was 37% which corresponded to 21.6% and 15.4% for moderate and severe anxiety, respectively. Sociodemographic factors such as age group and academic year were significantly associated with anxiety, while those with higher self-esteem (r_s_ = −0.487), self-competence (r_s_ = −0.407), self-liking (r_s_ = −0.499), and self-efficacy (r_s_ = −0.245) had lower anxiety. Inversely, those who adopted emotion-focused (r_s_ = 0.130) and dysfunctional coping styles (r_s_ = 0.559) showed higher anxiety. The main predictors of anxiety were self-liking as a protective factor (aOR = 0.81) and dysfunctional coping as a risk factor (aOR = 1.16). Therefore, resilience building and inculcating positive coping strategies are imperative in equipping our budding healthcare providers to weather through future unforeseeable disasters.

## 1. Introduction

The novel coronavirus disease 2019 (COVID-19), also known as severe acute respiratory syndrome coronavirus 2 (SARS-CoV-2), was first discovered in Wuhan, the capital city of Hubei province in China [1,2]. It started its initial spread back in December 2019, with many of those who had contracted the virus reporting symptoms such as fever, cough and respiratory illness of unknown origin [3]. On 30 January 2020, World Health Organization (WHO) declared the incident a public health emergency of international concern (PHEIC) due to the novelty and uncertainty of how the virus will behave and how quickly it had spread to over 20 other countries within a short span of time [4]. As the outbreak continued to spread like wildfire across the globe, causing diseases and deaths, the WHO declared COVID-19 a pandemic on 11 March 2020 [2]. COVID-19 hence prompted multiple lockdowns and quarantines, which primarily affected education and employment, in particular medical students.

Globally, medical students were equally affected by the lockdown decree by their respective countries, where medical training and education were halted, and they were forced to change their learning and life trajectories [5,6,7]. They had their professional exams canceled, graduation delayed, and placements for training postponed, with some even having their career progression affected in terms of applications for postgraduate training [6]. Some may argue that clinical skills can be taught using simulated-based learning, but traditional bedside teaching remains the gold standard. Those staying in rural areas find remote learning even more challenging with limited access to resources and the internet [5]. Pakistan has reported a high rate of mental health disorders among medical students, with prevalences of anxiety and depression of about 50% and higher in those with underlying psychiatric histories [7]. Additionally, students in Malaysia are also reported to have learning loss due to ineffective interactions between teachers, lack of preparation from educators, and lack of support from family members [8].

This study explores how specific constructs of self-esteem, self-efficacy, and coping strategies may have effects on anxiety among medical students. Self-esteem can be defined as one’s belief in how well one is living up to the standards of value placed by society [9]. It can also be seen as one’s views, values and personal feelings towards self that can influence one’s reaction toward life events [10]. A study conducted among Vietnamese students showed students with low self-esteem had an increased risk of anxiety, depression and even suicidal ideations [11]. Self-efficacy, or sense of self, reflects a person’s beliefs in their own ability to achieve a desirable outcome [12]. Studies have shown that self-efficacy plays a significant role in reducing mental health [12,13] and is associated with higher levels of well-being in the cognition and mood domains [14]. This is especially important in the medical setting, where students are trained in a challenging and competitive environment, which determines their personal growth and development in the field of medicine [13].

Coping skills, on the other hand, are a set of cognitive and behavioral efforts used to combat external threats from the environment or the internal demands of oneself that threaten one’s well-being [15]. According to Lazarus and Folkman’s transactional theory, the constant appraising of one’s environment for threats generates emotional distress when faced with a difficult situation, which in turn initiates coping strategies to manage the emotions or attempt to dampen the stressor [16]. Coping strategies can be divided into three domains: problem-focused coping (PFC), emotion-focused coping (EFC) and the less useful dysfunctional or avoidant coping (DC) strategies [17]. Anxiety and depression predicted the use of more maladaptive coping strategies than those without [18].

Furthermore, the level of distress also plays a role in the formation of psychopathology, with studies showing higher stress levels in females [19,20]. Stress could also lower life satisfaction, with the reverse being true [14]. Besides, healthcare workers with close contact with COVID-19 patients and, interestingly, those with no children and those who coped by seeking social support were among those at risk [20]. Whereas having good psychological mindedness and flexibility [19] with a positive attitude towards stressful situations can be a boon in coping with stress levels and anxiety [20].

Currently, there are a handful of studies looking into self-esteem [10,11], self-efficacy [21,22] and coping skills [23,24] with anxiety in different contexts and population groups. However, to the author’s current knowledge, there are very few to no published studies exploring the interrelationship between all these traits, together with anxiety among medical students, particularly in the Malaysian setting, during the COVID-19 pandemic. This study could also serve to further understand the prevalence of anxiety, the association between socio-demographic factors, and how they associate with one another. It is also timely to study how this unprecedented pandemic could lead to psychopathologies and to understand some of the adaptive or maladaptive coping strategies used. As evidenced by previous research showing that the COVID-19 pandemic and its ramifications could affect the mental health of medical students [13], the author hopes to recommend stakeholders integrate effective psychological education and interventions, provide more mental health training and awareness to our medical students to better equip them mentally for any future unforeseeable disasters. Furthermore, psychological studies like this can help demystify stigma among healthcare providers and inculcate understanding and empathy in terms of the challenges faced by those crippled with mental illnesses.

## 2. Materials and Methods

An observational cross-sectional study design was used to examine the association between psychological factors (self-efficacy, self-esteem and coping skills) and anxiety among medical students from a Malaysian public university. A 2-proportion sample calculation was used to determine the sample size, taking into account a 10% dropout rate. A simple random sampling method was used to disseminate the printed questionnaires physically to the medical students from year 1 through year 5 from 27 January 2022 to 27 May 2022. The participants were informed regarding the purpose of the study, and once they gave their consent, they were screened for inclusion and exclusion criteria. A total of 371 medical students were recruited with a favorable response rate and no missing values.

### 2.1. Inclusion Criteria

All registered medical students from Year 1 through Year 5 of university;Informed consent was given by medical students to participate.

### 2.2. Exclusion Criteria

Medical students who withdrew their consent after signing;Medical students who did not complete their questionnaire.

A self-constructed questionnaire was used to gather information on sociodemographic backgrounds such as age, sex, ethnicity, religion, academic year, exposure to COVID-19 news for more than 30 min, knowing someone infected with COVID-19, living arrangement, weight, and height. Besides these, self-esteem was measured using the Rosenberg Self-Esteem Scale (RSES), and the General Self-Efficacy Scale (GSES) was used to look at their self-efficacy, coping skills were measured using the Brief Coping Orientation to Problem Experienced Inventory (Brief COPE) scale, and lastly, the General Anxiety Disorder 7 (GAD-7) scale was used to identify anxiety symptoms. This study was ethically approved by the Ethics Committee for Research Involving Human Subjects, Universiti Putra Malaysia (JKEUPM-2021-427).

### 2.3. Measurement Tools

#### 2.3.1. Rosenberg Self-Esteem Scale (RSES)

RSES assesses global self-worth (self-competence and self-liking) based on positive and negative feelings about oneself. It was created by Morris Rosenberg in 1965. It applies a 10-item scale which is self-reported, using a 4-point Likert scale format ranging from strongly disagree for 1 to strongly agree for 4. Certain items have reverse scoring, with 4 of the items having positive statements and 5 having negative statements, and a higher score is indicative of higher self-esteem [25]. Sinclair et al. have proposed 2 subgroups of RSES which were self-competence (SC) and self-liking (SL), where SC is a person’s instrumental value while SL is a person’s intrinsic value. It involved summing the first 5 responses for SC and the last 5 responses for SL [26]. Validity and reliability tests were done among Malaysian students where the questionnaire was divided into 2 factors, namely the ‘positive items’ and the ‘negative items’ with a Cronbach alpha of 0.77 and 0.62, respectively [27].

#### 2.3.2. General Self-Efficacy Scale (GSES)

GSES assesses general perceived self-efficacy with regard to predicting coping and adaptation ability in both daily activities and stressful events. It was created by authors Ralf Schwarzer and Matthias Jerusalem in 1995 with a 10-item, self-administered questionnaire on a 4-point Likert scale. The response format is 1 for not at all true; 2 for hardly true; 3 for moderately true; 4 for exactly true, and scores ranging from 10 to 40, where the higher the score, the more self-efficacious one is. It is designed for the adolescent and adult populations with good reliability of Cronbach’s alpha range from 0.76 to 0.90 [28].

#### 2.3.3. Brief COPE

Brief COPE measures effective and ineffective ways of dealing with adverse life events in a 28-item inventory simplified by Charles Carver. It uses a 4-point scale ranging from 0, “I have not been doing this at all,” to 3, “I have been doing this a lot,” and has 14 individual coping styles which are Self-distraction, Denial, Substance Use, Behavioral disengagement, Emotional Support, Venting, Humor, Acceptance, Self-Blame, Religion, Active Coping, Use of Instrumental Support, Positive Reframing, and Planning. These 14 factors can be classified into 3 groups which are problem-focused, emotion-focused, and avoidant or dysfunctional coping. The Cronbach alpha values for each group were 0.81, 0.75, and 0.68, respectively [29].

#### 2.3.4. GAD-7

GAD-7 is a clinical measure that identifies anxiety symptoms which include generalized anxiety disorder, panic disorder, social anxiety disorder, and post-traumatic stress disorder (PTSD); in the past 2 weeks based on a self-rated, 7-item questionnaire. It is scored on a 4-point scale from 0, which is not at all, 1 for several days, 2 for more than half the days, and 3 for nearly every day; a total score of 0–21 can be obtained with cut-off points 0–4, 5–9, 10–14 and 15–21 for minimal, mild, moderate, and severe anxiety respectively. A study in Germany used GAD-7 with a cut-off score of ≥10 (moderate to severe severity) as an indication of symptomatic anxiety [30]. The Cronbach alpha is 0.92 with good test-retest reliability of 0.83 [31,32].

### 2.4. Statistical Analyses

The data were analyzed using the statistical software Statistical Package for the Social Sciences (SPSS) version 26 (IBM Corp., Armonk, NY, USA). The computed data for sociodemographic factors will be shown as a percentage for categorical data, whereas median and interquartile range (IQR) were used to describe continuous data as they were not normally distributed. In addition, the Mann–Whitney U test and Kruskal–Wallis test were used to look at comparisons between sociodemographic factors and anxiety, while Spearman’s correlation coefficient was used to examine psychological factors and anxiety. Finally, univariate ordered logistic regression analysis was used to predict anxiety among medical students with their statistical significance set between the confidence interval (CI) of 95%. Those data that were significant from the univariate analysis were followed through with stepwise multiple-ordered logistic regression analysis to account for confounders. A *p*-value of less than 0.05 was considered statistically significant.

## 3. Results

A total of 371 medical students participated in this study, with all respondents fulfilling the inclusion criteria.

### 3.1. Prevalence of Anxiety with Sociodemographic and Psychological Factors

Table 1 showed anxiety among medical students based on the GAD-7 scale, which was categorized into four levels, namely minimal, mild, moderate and severe. From the data, 118 students (31.8%) were minimally anxious, while another 116 students (31.3%) reported having mild anxiety. Meanwhile, 80 students (21.6%) and 57 students (15.4%) showed moderate and severe levels of anxiety, respectively. Furthermore, more than one-third of students (N = 137, 37%) scored 10 or more, which was indicative of symptomatic anxiety, while the remainder, 234 students (63%), did not.

Out of 371 medical students that participated, the majority of them were between the ages of 23–26 years old (55%) in the older age group, while the others fall between 19 to 22 years old (45%) which were in the younger age group. The younger age group showed a higher level of severe anxiety (57.9%) as compared to the older age group (42.1%). Moreover, there were more female medical students at 66.6% as compared to their male counterparts at only 33.4%, whereas the female students showed higher levels of severe anxiety at 71.9%. Most of them who joined the study were of Malay descent (52.8%), followed by Indian descent (23.5%), Chinese descent (22.1%), and others (1.6%), with students of Malay descent, predominated the severe anxiety level at 52.6%. Religion-wise, most of the students were practicing Islam (55.5%), followed by Hinduism (20.2%), Buddhism (16.7%), Christianity (6.5%), and others (1.1%). Muslim students showed higher levels of severe anxiety (56.1%) as compared to students of other religions. Furthermore, more participants were in Year 1, with 85 students (22.9%), 76 (20.5%) in Year 2, 78 (21%) in Year 3, 52 (14%) in Year 4, and 80 (21.6%) in their final year. Overall, students in Year 1 (N = 38) had more symptomatic anxiety (≥10 in GAD-7) as compared to those in other academic years. Interestingly, a bulk of the students (69%) did not spend much time (less than 30 min) reading COVID-19 news in a day, whereas only 31% of them did so (more than 30 min a day). The result showed those who did not spend much time on COVID-19 news had higher levels of severe anxiety (64.9%) as compared to those who did (35.1%). Almost all of the students were staying with others, including families (96.5%), while only 3.5% were staying alone. There were 234 students (63.1%) who immediately knew someone who was infected with the COVID-19 virus, and 137 students (36.9%) knew someone distant. Those who had close ones infected with COVID showed more severe levels of anxiety (77.2%) as compared to those who did not (36.9%). Lastly, the majority of the students fall within the healthy weight range (63.3%), and the others were 16.2% for underweight, 13.2% and 7.3% for overweight and obese, respectively.

Table 2 shows the psychological factors among medical students, whereby among them, self-esteem had a median of 18.0 with an IQR of 6.0. This meant after arranging the data from least to most, half of the students at point 18 had lesser self-esteem, and the other half had more. The categorization of self-esteem into SC and SL both gave a median and IQR of 9.0 (3.0). Besides, half of the students at point 29.0 (5.0) had lower self-efficacy, and the others were more self-efficacious. Coping-wise, half of 28.0 (7.0) and 17.0 (4.0) used less EFC and PFC, respectively, while the other half used more for both. DC, on the other hand, has a median of 24.0 with an IQR of 8.0.

### 3.2. Association and Correlation Analysis of Sociodemographic and Psychological Factors with Anxiety

Table 3 shows the association between sociodemographic factors and anxiety among medical students. From the result, only age group (*p* = 0.006, U = 14335, Z = −2.733) and academic year (*p* = 0.001, $x^{2}$ = 19.510) were statistically significant with a *p*-value less than our chosen significance level (α = 0.05). We can thus conclude that both age group and academic year have an association with anxiety among medical students with academic year disruption during the COVID-19 pandemic. Meanwhile, other factors such as gender (*p* = 0.306), COVID-19 news time of more than 30 min per day (*p* = 0.452), living arrangement (*p* = 0.816), knowing someone infected (*p* = 0.360), ethnicity (*p* = 0.714), religion (*p* = 0.686), and BMI group (*p* = 0.386), showed no association between these factors with anxiety among medical students.

Table 4 shows the result of Spearman’s rank correlation between psychological factors and anxiety among medical students. Self-esteem (r_s_ = −0.487, *p* ≤ 0.001) and its subcategories of SC (r_s_ = −0.407, *p* ≤ 0.001) and SL (r_s_ = −0.499, *p* ≤ 0.001) all showed statistical significance and had negative correlation towards anxiety. In addition, self-efficacy also showed a negative correlation towards anxiety with statistical significance (r_s_= −0.245, *p* = <0.001). Whereas, among the coping styles, EFC (r_s_ = 0.130, *p* = 0.012) and DC (r_s_ = 0.559, *p* ≤ 0.001) both showed a positive correlation towards anxiety and were statistically significant while PFC did not yield a statistically significant result (*p* = 0.141). This implied that those with higher self-esteem (both in self-competence and self-liking) and who were self-efficacious had an inverse effect on anxiety. Contrary to that, those who adopted a more emotion-focused and dysfunctional type of coping would have higher anxiety.

### 3.3. Predictors of Anxiety using Univariate and Multivariate Ordered Logistics Regression

Based on Table 5, seven variables were found to be statistically significant, with a *p*-value of less than 0.05 in the univariate analyses. The variables were age group, academic year, self-esteem and its categories SC and SL, self-efficacy and DC, which showed a relationship between them and anxiety among medical students. Those in the younger age group were 1.68 (95% CI = 1.158, 2.435) times more likely to have anxiety, whereas those in the academic year 4 had an OR of 0.33 (95% CI = 0.170, 0.627). Other than that, having high self-esteem (OR = 0.82, 95% CI = 0.783, 0.854), SC (OR = 0.73, 95% CI = 0.675, 0.788), SL (OR = 0.68, 95% CI = 0.628, 0.737) and self-efficacy (OR = 0.89, 95% CI = 0.850, 0.928) were less likely to have anxiety with a crude OR of less than 1. Those using DC, on the other hand, were 1.22 (95% CI = 1.174, 1.273) times higher to develop anxiety.

A subsequent multicollinearity analysis was done to identify the Variance Inflation Factors (VIF) score for a strong correlation between variables. Both age group and academic year had high VIF scores of 14.45 and 14.51, respectively, which showed a strong correlation between the variables and were excluded in the multivariate analysis along with other factors that had *p*-values of more than 0.05. Self-esteem, as a main variable, was also excluded as the categories of SC and SL were analyzed instead. The variables SC, SL, self-efficacy and DC all had VIF scores of less than 5 and were included in the multivariate analysis.

From the multivariate analyses, only SL and DC were statistically significant, with a *p-*value of less than 0.05. Those with higher self-liking traits were less likely to have anxiety with an adjusted odds ratio (aOR) of 0.81 (95% CI: 0.716, 0.912), while those who adopted DC were 1.16 (95% CI: 1.112, 1.211) times more likely to develop anxiety. These 2 variables were likely to predict anxiety among medical students.

## 4. Discussion

A meta-analysis revealed a global prevalence of anxiety among medical students, which was around 33.8%, prior to the COVID-19 pandemic [33]. In our study, an increase in the prevalence of anxiety of 37% was seen among medical students in a tertiary education center during the COVID-19 pandemic, with a cutoff point of more than 10 on the GAD-7 scale. Of these, 21.6% had moderate anxiety, while 15.4% had severe anxiety. This finding was almost similar to an Iranian study done among medical students, with a prevalence of 38% [34]. The identifiable reason for our numbers to be slightly lower than previous studies was perhaps due to the period of data collection, which was taken early to middle of the year 2022, which was towards the end of the pandemic, where the restrictions of movement control order (MCO) were lessened, and students were somewhat able to adapt to the new norm. However, the anxiety levels are still higher than the prevalence outside the pandemic era in a comparable group of students from the same population [35]. This shows that despite the resilience of the students leading to lesser anxieties in 2022 compared to 2020, the residual COVID-19 anxieties are still gripping students and need to be intervened upon. The ability to adapt can reduce the initial anxiety experienced by students, as seen in those who have contracted COVID-19 [34]. Our study also found that those in the younger age group of 19–22 years (cut-off point of more than 10 on the GAD-7 scale, N = 72), in particular newly enrolled medical students in Year 1 (N = 38), had significantly higher anxiety level as compared to their seniors. The underlying mechanism may be that anxiety is a response to the uncertainty of changes and stems from the sympathetic fight or flight response when change is imminent and is deemed by the individual to be out of control [36]. They were more likely to be uncertain about their study program, multiple adjustments after joining medical school during the pandemic and lesser maturity and coping abilities to face challenges. A study also showed students in their pre-clinical years reported a higher prevalence of anxiety attributed to relocation, distractions at home and less experience in medical school [37]. Moreover, another study reported younger students having higher anxiety attributed to high social media use; however, the study population was generalized to other fields of study as well [38].

Our study did not yield statistical significance for gender, ethnicity, religion, living arrangements, COVID-19 news time, immediately knowing someone with COVID-19 and BMI. This differs somewhat from previous literature, where the female gender is generally a predictor for psychological distress, especially anxiety and health anxiety [39]. However, medical students may be a more homogenous group compared to a random population-based sampling of males and females, and hence the gender differences may not be as evident as male and female medical students alike are undergoing the same stressors. In addition, more than half of the participants were of Malay ethnicity and practiced Islam as their religion, more than 95% of the students were living with others, either with family or friends, and around 60% had normal BMI. This disparity in sample size could make interpreting the outcome challenging.

Browning et al. studied BMI as a potential risk factor for having a psychological impact from the pandemic; however, it did not show statistical significance [40], similar to our study. Interestingly, around 70% of the students from our study had COVID-19 news time of less than 30 min a day, possibly due to online platforms reporting the same news repeatedly, and they were only keeping track of the total numbers infected on a daily basis which required no time at all. There were a few studies that reported increased screen time with regards to the COVID-19 pandemic to be a determinant for anxiety [41], from exhaustion by overwhelming negative information [38], and from increased insecurity, stigma and economic downturn [42]. Finally, around two-thirds of students knew someone close to them infected by the virus. The likelihood of having anxiety increases by 1.6 times if close ones are infected by COVID-19 [41].

The median and IQR score of self-esteem was 18.0 (6.0), and both SC and SL were 9.0 (3.0). They scored lower than an international study, with 53 nations participating, where Malaysia scored a mean of 29.83 ± 3.42 for self-esteem, 16.42 ± 1.99 for SC and 13.38 ± 2.01 for SL [43]. This could be reflected by the different population groups studied and also during the pre-COVID-19 pandemic. Apart from that, our study revealed a significant association between psychological factors of self-esteem and its subcategories (SC and SL) with anxiety. These determinants showed a negative correlation between anxiety with higher self-esteem (SC and SL), resulting in reduced anxiety among medical students. The results were in line with a study from Japan among medical students during the pandemic, whereby students with higher scores in RSES showed lower psychological distress [13]. Those who were younger, being in social isolation and of the female gender had lower SL, which had an affective notion, with the reverse being true [26]. The age factor was similarly significant in our study, showing that younger people had lower self-esteem, in particular SL (after adjustments in logistic regression), which is a protective factor against anxiety. A study from Pakistan also showed self-esteem as a predictor of social connectedness and inversely predicted social anxiety [10]. Furthermore, some medical students did not join the medical course out of their own accord but instead to either fulfill their role as a filial child due to vicarious parenting or from the socially perceived high status of becoming a doctor. This added burden could render them with low academic performances, which could also impact their self-esteem among peers, reduce their perceived competence, and fuel their anxiety [44].

Following that, self-efficacy drives one to perform new and challenging tasks and motivates them to reach the end goal. This is because self-efficacy is responsible for the formation of a sustainable attitude and shrinking the attitude-behavior gap. Hence, Kornilaki et al. postulate that it partly shifts the locus of responsibility for an inability to act sustainably toward the situation, which is modifiable, rather than the individual in question [45]. The median and IQR of self-efficacy was 29.0 (5.0), which was higher compared to a Chinese study done among nurses during the pandemic. However, the Chinese study was done at an earlier time in the pandemic when there was more chaos and less certainty regarding the course of the pandemic. Hence, medical students in this study frame have had two years to develop resilience and cope with the increasingly predictable stressors of the pandemic, and their self-efficacy might be consequently higher. This could also be explained that due to the time differences, medical staff were not as equipped knowledge-wise about the pandemic then as they were now, thus showing a difference in their confidence and motivation. The same study also reflected a similar result to ours, which showed a negative correlation between self-efficacy and anxiety [46]. The uncertainties of the pandemic hindered one’s ability to perform to their fullest, causing hesitancies and anxiety. However, with knowledge about the unknown, we learn to cope and adapt, which in turn reduces potential anxiety. Moreover, having high self-efficacy also positively correlates with assertiveness which could negatively predict anxiety [12].

Studies have also suggested the use of psychotherapy, such as cognitive-behavioral therapy (CBT) and mindfulness-based interventions, to improve students’ self-esteem by targeting the SL facet [26] and increasing self-efficacy [46]. On the other hand, incorporating activities and curriculum that strengthen both self-esteem and self-efficacy, providing useful resources to detect students that are struggling and pave the way for educational interventions among them, could be useful in strengthening these psychological domains [13].

Besides that, out of the three categories of Brief COPE, EFC, and DC were statistically significant in the study showing a positive correlation with anxiety. EFC was used the most by students with a median and IQR of 29.0 (7.0), followed by DC at 24.0 (8.0), and the least used is the adaptive PFC at only 17.0 (4.0). According to Cooper et al., the sub-domains of EFC were using emotional support, positive reframing, acceptance, religion and humor; while DC included venting, denial, substance use, behavioral disengagement, self-distraction and self-blame; PFC, on the other hand, included active coping, planning and use of instrumental support [47]. Students who adopt EFC mainly deal with problems indirectly by negating unfavorable emotions, while those who actively seek out solutions to solve the problems use PFC, which is the best coping strategy there is [42]. A study showed those who use EFC, particularly those seeking social and emotional support, have a positive correlation with perceived stress which is associated with anxiety. The frustration stemmed from the inability to socially confide with their families as they fear the risk of infection [20]. This was similar to our finding where students might feel defeated with their futile efforts to cope emotionally as the stressor is overwhelmingly out of their control. Additionally, our study showed DC to have a positive correlation with anxiety which echoed a few previous studies [42,48]. DC was also found to be a risk factor for anxiety in the regression analysis model, which was similar to a previous study done as well [49]. Those with a higher predisposition to anxiety and COVID-19-related stressors prefer to vent out their problems instead of dealing with them, constantly use denial as a form of defense, would personalize problems and blame themselves when the outcome is unfavorable, adopts excessive use of self-distraction and behavioral disengagement as means to avoid overwhelming the psyche. Moreover, they would also seek out substances to dampen their problems, creating more public health epidemics. 4% of students in Nigeria reported the use of drugs during the COVID-19 lockdown citing for relaxation and boredom, possibly from increased anxiety due to the quarantine [50]. Although focusing on the more adaptive ways through PFC to self-regulate emotionally would increase the students’ resilience to adversities and reduce anxiety, the need to identify and address potential maladaptive coping strategies also has to be set in place. Changes need to occur at the grassroot level with teachers encouraging students to increase their autonomy by planning, taking charge of their course of action and seeking out support from others when needed [15].

## 5. Limitations

Several limitations were identified in this study. Firstly, we only took samples from one public medical school, which could not fully represent the diverse sociodemographic background and psychological resilience among medical students (both in private and in public) in Malaysia from the impact of the COVID-19 pandemic. Furthermore, the nature of the study design of cross-sectional meant that the result could not yield causation but only look at associations between variables. Additionally, the data collection was done towards the end of the pandemic, thus could not truly reflect the true anxiety of what the students went through during the height of the pandemic. Moreover, there are a few potential limitations to our measures. Firstly, the nature of the questionnaires, which were self-rated, could mean that some level of bias was present. Secondly, as cut-off points for the scales employed were established prior to the COVID-19 pandemic, the cut-off points in the current study may be an inaccurate estimate of the true prevalence of the illnesses screened for, hence further separate studies establishing ROC curves and pandemic-adjusted optimal sensitivities and specificities can be performed in subsequent projects.

## 6. Conclusions

The COVID-19 pandemic was shown to create waves of impact on the mental well-being of medical students across the globe. The vulnerability of students in the younger age groups in dealing with this unprecedented disaster would be insurmountable without the proper mental preparation and psychological resilience equipped. Factors such as self-esteem with facets of SC and SL and self-efficacy were associated with anxiety which proved that efforts at the grassroots level are required to inculcate these psychological constructs in our budding healthcare providers. Echoing that, supervisors should also be vigilant in identifying any dysfunctional coping adopted by their students as these maladaptive behaviors would only protect the psyche momentarily but have devastating effects in the long run, which could perpetuate not only anxiety but other mental health conditions as well. With the proper primary preventive measures and appropriate interventions, we can ensure that our students are able to weather through any future unforeseeable disasters.

## Figures and Tables

**Table 1 ijerph-20-01894-t001:** Sociodemographic Factors and Level of Anxiety among medical students.

Variables	Level of Anxiety				Total
	Minimal	Mild	Moderate	Severe	
	N = 118 (31.8%)	N = 116 (31.3%)	N = 80 (21.6%)	N = 57 (15.4%)	N = 371 (100%)
Age Group					
19–22 (Younger group)	43 (36.4)	52 (44.8)	39 (48.8)	33 (57.9)	167 (45.0)
23–26 (Older group)	75 (63.6)	64 (55.2)	41 (51.2)	24 (42.1)	204 (55.0)
Gender					
Male	41 (34.7)	43 (37.1)	24 (30.0)	16 (28.1)	124 (33.4)
Female	77 (65.3)	73 (62.9)	56 (70.0)	41 (71.9)	247 (66.6)
Ethnic					
Malay	61 (51.7)	63 (54.3)	42 (52.5)	30 (52.6)	196 (52.8)
Chinese	27 (22.9)	30 (25.9)	13 (16.2)	12 (21.1)	82 (22.1)
Indian	27 (22.9)	22 (19.0)	24 (30.0)	14 (24.6)	87 (23.5)
Others	3 (2.5)	1 (0.9)	1 (1.2)	1 (1.8)	6 (1.6)
Religion					
Islam	67 (56.8)	63 (54.3)	44 (55.0)	32 (56.1)	206 (55.5)
Buddhist	20 (16.9)	24 (20.7)	9 (11.2)	9 (15.8)	62 (16.7)
Hindu	22 (18.6)	20 (17.2)	20 (25.0)	13 (22.8)	75 (20.2)
Christian	9 (7.6)	7 (6.0)	5 (6.2)	3 (5.3)	24 (6.5)
Others		2 (1.7)	2 (2.5)		4 (1.1)
Academic Year					
Year 1	22 (18.6)	25 (21.6)	23 (28.7)	15 (26.3)	85 (22.9)
Year 2	19 (16.1)	25 (21.6)	15 (18.8)	17 (29.8)	76 (20.5)
Year 3	28 (23.7)	20 (17.2)	18 (22.5)	12 (21.1)	78 (21.0)
Year 4	29 (24.6)	14 (12.1)	7 (8.8)	2 (3.5)	52 (14.0)
Year 5	20 (16.9)	32 (27.6)	17 (21.2)	11 (19.3)	80 (21.6)
COVID-19 > 30 min					
No	83 (70.3)	82 (70.7)	54 (67.5)	37 (64.9)	256 (69.0)
Yes	35 (29.7)	34 (29.3)	26 (32.5)	20 (35.1)	115 (31.0)
Living Arrangement					
Living alone	5 (4.2)	2 (1.7)	3 (3.8)	3 (5.3)	13 (3.5)
Living with others	113 (95.8)	114 (98.3)	77 (96.2)	54 (94.7)	358 (96.5)
Knowing someone infected					
Someone immediate	78 (66.1)	62 (53.4)	50 (62.5)	44 (77.2)	234 (63.1)
Someone distant	40 (33.9)	54 (46.6)	30 (37.5)	13 (22.8)	137 (36.9)
BMI Group					
Underweight	18 (15.3)	18 (15.5)	15 (18.8)	9 (15.8)	60 (16.2)
Healthy weight	79 (66.9)	74 (63.8)	46 (57.5)	36 (63.2)	235 (63.3)
Overweight	10 (8.5)	18 (15.5)	12 (15.0)	9 (15.8)	49 (13.2)
Obese	11 (9.3)	6 (5.2)	7 (8.8)	3 (5.3)	27 (7.3)

**Table 2 ijerph-20-01894-t002:** Psychological Factors among Medical Students.

Psychological Factors	Median (IQR)
Self-Esteem	18.0 (6.0)
Self-Competence (SC)	9.0 (3.0)
Self-Liking (SL)	9.0 (3.0)
Self-Efficacy	29.0 (5.0)
Emotion-Focused Coping (EFC)	28.0 (7.0)
Problem-Focused Coping (PFC)	17.0 (4.0)
Dysfunctional Coping (DC)	24.0 (8.0)

IQR: Interquartile range.

**Table 3 ijerph-20-01894-t003:** Association between Sociodemographic Factors and Anxiety among Medical Students.

Sociodemographic Factors	Mean Rank	U	Z	ChiSquare *x*^2^	*p*-Value
Age Group					
19–22 (younger age group)	202.16	14335.000	−2.733		0.006 *
23–26 (older age group)	172.77				
Gender					
Male	178.27	14355.000	−1.024		0.306
Female	189.88				
COVID-19 > 30 min					
No	183.30	14029.500	−0.752		0.452
Yes	192.00				
Living Arrangement					
Living alone	192.54	2242.000	−0.233		0.816
Living with others	185.76				
Knowing Someone Infected					
Someone Immediate	189.75	15152.000	−0.915		0.360
Someone Distant	179.60				
Ethnic					
Malay	186.57			1.365	0.714
Chinese	177.88				
Indian	194.02				
Others	162.08				
Religion					
Islam	185.24			2.270	0.686
Buddhist	177.15				
Hindu	197.17				
Christian	173.85				
Others	225.50				
Academic Year					
Year 1	202.12			19.510	0.001 *
Year 2	203.84				
Year 3	182.73				
Year 4	130.85				
Year 5	190.97				
BMI Group					
Underweight	190.88			3.035	0.386
Healthy Weight	181.86				
Overweight	207.20				
Obese	172.74				

U: Mann Whitney U test; Z: Z test; *: *p* ≤ 0.05.

**Table 4 ijerph-20-01894-t004:** Correlation between Psychological Factors and Anxiety among Medical Students.

Anxiety and Self-Esteem			
		Anxiety	Self-Esteem
Anxiety	Correlation coefficient	1.000	−0.487
	Sig. (two-tailed)		0.000 *
	N	371	371
Self-Esteem	Correlation coefficient	−0.487	1.000
	Sig. (two-tailed)	0.000 *	
	N	371	371
Anxiety and SC			
		Anxiety	SC
Anxiety	Correlation coefficient	1.000	−0.407
	Sig. (two-tailed)		0.000 *
	N	371	371
SC	Correlation coefficient	−0.407	1.000
	Sig. (two-tailed)	0.000 *	
	N	371	371
Anxiety and SL			
		Anxiety	SL
Anxiety	Correlation coefficient	1.000	−0.499
	Sig. (two-tailed)		0.000 *
	N	371	371
SL	Correlation coefficient	−0.499	1.000
	Sig. (two-tailed)	0.000 *	
	N	371	371
Anxiety and Self-Efficacy			
		Anxiety	Self-Efficiency
Anxiety	Correlation coefficient	1.000	−0.245
	Sig. (two-tailed)		0.000 *
	N	371	371
Self-Efficiency	Correlation coefficient	−0.245	1.000
	Sig. (two-tailed)	0.000 *	
	N	371	371
Anxiety and EFC			
		Anxiety	EFC
Anxiety	Correlation coefficient	1.000	0.130
	Sig. (two-tailed)		0.012 *
	N	371	371
EFC	Correlation coefficient	0.130	1.000
	Sig. (two-tailed)	0.012 *	
	N	371	371
Anxiety and PFC			
		Anxiety	PFC
Anxiety	Correlation coefficient	1.000	0.077
	Sig. (two-tailed)		0.141
	N	371	371
PFC	Correlation coefficient	0.077	1.000
	Sig. (two-tailed)	0.141	
	N	371	371
**Anxiety and DC**			
		**Anxiety**	**DC**
**Anxiety**	Correlation coefficient	1.000	0.559
	Sig. (two-tailed)		0.000 *
	N	371	371
**DC**	Correlation coefficient	0.559	1.000
	Sig. (two-tailed)	0.000 *	
	N	371	371

*: Correlation is significant at the 0.05 level (2-tailed); Sig: Significance; N: Number of students; SC: Self-Competence, SL: Self-Liking, EFC: Emotion Focused Coping; PFC: Problem Focused Coping; DC: Dysfunctional Coping.

**Table 5 ijerph-20-01894-t005:** Multivariate Ordered Logistics Regression Analysis between Variables and Anxiety among Medical Students.

Variables	Crude OR (95% CI)	*p*-Value	Adjusted OR (95% CI)	*p-*Value
Age Group				
19–22 (younger age group)	1.679 (1.158–2.435)	0.006 *		
23–26 (older age group)				
Gender				
Male	1.224 (0.830–1.803)	0.310		
Female				
Ethnicity				
Malay	1.650 (0.335–8.134)	0.511		
Chinese	1.419 (0.280–7.186)	0.653		
Indian	1.884 (0.372–9.532)	0.414		
Others				
Religion				
Islam	0.592 (0.129–2.722)	0.562		
Buddhist	0.515 (0.107–2.478)	0.473		
Hindu	0.731 (0.154–3.480)	0.733		
Christian	0.483 (0.090–2.575)	0.453		
Others				
Academic Year				
Year 1	1.225 (0.713–2.104)	0.469		
Year 2	1.270 (0.725–2.225)	0.407		
Year 3	0.867 (0.494–1.521)	0.619		
Year 4	0.326 (0.170–0.627)	0.001 *		
Year 5				
COVID-19 news time > 30 min				
No	0.858 (0.576–1.277)	0.448		
Yes				
Living Arrangement				
Living Alone	0.146 (0.395–3.323)	0.789		
Living with Others				
Knowing someone infected				
Someone Immediate	1.189 (0.817–1.731)	0.373		
Someone Distant				
BMI Group				
Underweight	1.388 (0.603–3.196)	0.435		
Healthy Weight	1.185 (0.567–2.476)	0.646		
Overweight	1.813 (0.772–4.261)	0.170		
Obese				
Self Esteem	0.817 (0.783–0.854)	<0.001 *		
SC	0.729 (0.675–0.788)	<0.001 *	0.923 (0.819–1.041)	0.192
SL	0.680 (0.628–0.737)	<0.001 *	0.808 (0.716–0.912)	0.001 *
Self-Efficiency	0.888 (0.850–0.928)	<0.001 *	1.014 (0.959–1.072)	0.618
EFC	1.034 (0.998–1.072)	0.065		
PFC	1.031 (0.976–1.089)	0.282		
DC	1.223 (1.174–1.273)	<0.001 *	1.160 (1.112–1.211)	<0.001 *

*: *p* ≤ 0.05; OR: Odds ratio; CI: Confidence interval; SC: Self-Competence, SL: Self-Liking, EFC: Emotion Focused Coping; PFC: Problem Focused Coping; DC: Dysfunctional Coping.

## Data Availability

The data in this study are available upon request from the corresponding author and are not publicly available due to confidentiality.

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
