# Peer review of "Psychological Resilience and Coping Strategies with Anxiety among Malaysian Medical Students during the COVID-19 Pandemic"

_ijerph, 2023, doi:10.3390/ijerph20031894_

Round 1
Reviewer 1 Report
Psychological Resilience and Coping Strategies with Anxiety 2 Among Malaysian Medical Students During the COVID-19 3 Pandemic
I have reviewed the manuscript, titled “Psychological Resilience and Coping Strategies with Anxiety Among Malaysian Medical Students During the COVID-19 3 Pandemic”. The study was to investigate the association between sociodemographic factors, psychological factors, coping strategies and anxiety among medical students as well as to identify the predictors of anxiety amongst them. A cross-sectional study design was used.
The study has some strong points (clear introduction, accurate statistical analysis, constructive discussion).
However, I would like to ask the authors to address some points in order to improve the paper
Introduction:
1) The study examines the impact of COVID-19 pandemic on psychological factors, coping strategies and anxiety among medical students. Therefore, it would be appropriate to provide more information on the role of stress during the pandemic (see: Krok, D., Zarzycka, B., & Telka, E. (2021). Risk of contracting COVID-19, personal resources and subjective well-being among healthcare workers: the mediating role of stress and meaning-making. Journal of Clinical Medicine, 10(1), 132; Babore, A., Lombardi, L., Viceconti, M. L., Pignataro, S., Marino, V., Crudele, M., ... & Trumello, C. (2020). Psychological effects of the COVID-2019 pandemic: Perceived stress and coping strategies among healthcare professionals. Psychiatry research, 293, 113366)
2) Can you present some research on relationships between anxiety and coping?.
Method:
3) Was the sample determined by power analysis?
4) How did you handle missing values in your data? (If any exist)
Discussion:
5) What are the underlying mechanisms responsible for this result: “the younger age group of 19 to 22 years (cut off point of more than 10 in GAD-7 scale, N=72), in particular newly enrolled medical students in Year 1 (N=38) had significantly higher anxiety level as compared to their seniors” (p. 13)?
6) Can you elaborate on the following statement: “Self-efficacy drives one to perform new and challenging tasks and motivates them to reach the end goal (p. 14).”. Please, provide a potential explanation of this result.
7) Is there any limitation related to the measures used in your study?
Reviewer 2 Report
1. Abstract: Self-liking is a protective factor, while coping strategy dysfunction is a risk factor. That's a better description than just using a predictor.
2. Introduction: There are so many descriptions of the spread and infection of COVID-19 that there is no need to explain the information as it is well known.
The survey was conducted in 2022, when only omicron was circulating globally and the COVID-19 fatality rate dropped significantly. Therefore, death theory is not suitable to explain the anxiety in the introduction.
This section needs to be simplified.
3. Methods: How to collect these participants? How are the questionnaires distributed?
Continuous variables should be shown in mean plus or minus SD or IQR, depending on the distribution.
4. Results: Tables 1, 2 and 4 need to be combined and simplified.
A subgroup analysis should be conducted for students infected and uninfected with COVID-19. Due to different psychological responses to COVID-19, for example, infected students may worry about discrimination from others, while uninfected students may worry about exposure to COVID-19.
5. Discussion: "the period of data collection which was taken early to 325 middle of the year 2022 which was towards the end of the pandemic, where the re-326 strictions of movement control order (MCO) were lessened and students were somewhat 327 able to adapt to the new norm" This paragraph is inconsistent with the introduction which describes many bad environments and negative effects on students.
This phenomenon of low anxiety among Chinese and American medical students has not been discussed in depth, nor has its relationship with this study been revealed.
The discussion was not in-depth and appropriate, for example, the self-efficacy results obtained in the medical student study were not comparable to those obtained in the nurse study.
Inconsistencies with other studies are not fully discussed.
Round 2
Reviewer 2 Report
There is no final conclusion on whether the novel coronavirus really originated in Wuhan, China, and the World Health Organization also believes that the true source tracing is impossible. Thus the first sentence of Introduction need to be revised, the literatures quoted in the first sentence are also not authoritative journals, so it is suggested to be modified to describe that the novel coronavirus was first discovered in Wuhan, which is more rigorous.
